# Characterisation and Expression Analysis of LdSERK1, a Somatic Embryogenesis Gene in *Lilium davidii* var. *unicolor*

**DOI:** 10.3390/plants13111495

**Published:** 2024-05-29

**Authors:** Shaojuan Wang, Xiaoyan Yi, Lijuan Zhang, Muhammad Moaaz Ali, Mingli Ke, Yuxian Lu, Yiping Zheng, Xuanmei Cai, Shaozhong Fang, Jian Wu, Zhimin Lin, Faxing Chen

**Affiliations:** 1College of Horticulture, Fujian Agriculture and Forestry University, Fuzhou 350002, China; 3210330057@fafu.edu.cn (S.W.); 1210305019@fafu.edu.cn (X.Y.); 1210305020@fafu.edu.cn (L.Z.); moaaz@fafu.edu.cn (M.M.A.); 5220330046@fafu.edu.cn (M.K.); 5220330020@fafu.edu.cn (Y.L.); 2Fujian Academy of Agricultural Sciences Biotechnology Institute, Fuzhou 350003, China; zhengyiping@faas.cn (Y.Z.); caixuanmei@faas.cn (X.C.); fangshaozhong@faas.cn (S.F.); 3Beijing Key Laboratory of Development and Quality Control of Ornamental Crops, Department of Ornamental Horticulture, China Agricultural University, Beijing 100193, China; jianwu@cau.edu.cn

**Keywords:** gene isolation, somatic embryogenesis receptor-like kinase (SERK) gene, *LdSERK1*, phenotyping, *Lilium davidii* var. *unicolor*

## Abstract

The Lanzhou lily (*Lilium davidii* var. *unicolor*) is a variant of the Sichuan lily of the lily family and is a unique Chinese ‘medicinal and food’ sweet lily. Somatic cell embryogenesis of *Lilium* has played an important role in providing technical support for germplasm conservation, bulb propagation and improvement of genetic traits. *Somatic embryogenesis receptor-like kinases* (SERKs) are widely distributed in plants and have been shown to play multiple roles in plant life, including growth and development, somatic embryogenesis and hormone induction. Integrating the results of KEGG enrichment, GO annotation and gene expression analysis, a lily LdSERK1 gene was cloned. The full-length open reading frame of *LdSERK1* was 1875 bp, encoding 624 amino acids. The results of the phylogenetic tree analysis showed that *LdSERK1* was highly similar to rice, maize and other plant SERKs. The results of the subcellular localisation in the onion epidermis suggested that the *LdSERK1* protein was localised at the cell membrane. Secondly, we established the virus-induced gene-silencing (VIGS) system in lily scales, and the results of *LdSERK1* silencing by Tobacco rattle virus (TRV) showed that, with the down-regulation of *LdSERK1* expression, the occurrence of somatic embryogenesis and callus tissue induction in scales was significantly reduced. Finally, molecular assays from overexpression of the *LdSERK1* gene in *Arabidopsis* showed that *LdSERK1* expression was significantly enhanced in the three transgenic lines compared to the wild type, and that the probability of inducing callus tissue in seed was significantly higher than that of the wild type at a concentration of 2 mg/L 2,4-D, which was manifested by an increase in the granularity of the callus tissue.

## 1. Introduction

Somatic embryogenesis is a widespread phenomenon in plants, including cereals, beans and nuts, which is why land plants are often called embryophytes [1]. The individual development of plant embryos can be divided into different stages, from fertilisation to maturity [2]. Somatic embryogenesis redifferentiates according to the embryonic programme, including the establishment of the somatic axis (polarity) and the primary and terminal meristematic tissues [3]. After fertilisation in *Arabidopsis*, the nucleus of the fertilised egg moves to the apical pole, and the fertilised egg divides asymmetrically to produce a small apical cell, giving rise to the whole embryo [4]. The embryo is multicellular, consisting of a large number of diploid cells produced by mitosis of the fertilised egg. Continuous cell division is therefore an essential part of embryogenesis [5]. An embryo is the product of fertilisation, formed by the fusion of cells from the organs of the maternal and paternal parents [6]. As early as the late 19th century, the ability of plants to regenerate attracted the interest of the scientific community [7]. Studies have shown that cytokinins and growth hormones lead to the proliferation of cell clusters that resemble wound-healing plant tissues, collectively known as ‘callus tissue’ [8]. Somatic embryogenesis is a natural phenomenon in which somatic embryos are formed from somatic cells. It is considered to be the most efficient morphogenetic pathway for plant reproduction [9]. It has been shown that, in addition to continuous aboveground and root organogenesis, plants can be regenerated from cultured plant cells by embryogenesis, and this pathway has been termed ‘somatic embryogenesis’ [10]. The potential for somatic embryogenesis depends on the genotype, the composition of the explant and culture medium, especially the plant growth regulators, and the environmental conditions. Typically, the process consists of three stages: induction of embryonic callus tissue, proliferation of embryonic callus cells and germination of somatic embryos [11]. Somatic embryogenesis involves the action of complex signalling networks, and this regulation is usually in response to exogenous stimuli generated by plant growth regulators or specific stress conditions, mainly low or high temperature, osmotic shock, heavy metals or drought [12]. The type of explant, the age and genotype of the mother plant, the physiological conditions of the culture, the cell density of the suspension culture and the production of homogeneous cell aggregates are all factors that must be taken into account in order to generate embryogenic potential [13]. In addition, the process by which plant embryonic callus tissues undergoes somatic embryogenesis to regenerate plants is mediated by regulatory factors such as transcription factors and specifically expressed genes [14].

To date, most studies have investigated the mechanisms of SE process induction using model plant species such as *Arabidopsis thaliana* [15], carrot [16], alfalfa [17], maize [18] and rice [19]. Lilies are monocotyledons of the *Liliaceae* and are the most important bulb crops [20]. Due to the lack of genomic information on lilies, early research on SE focused on the use of plant growth regulators and the selection of explants [21]. Somatic cell embryogenesis was first reported in some *Lilium* hybrids [22]. Subsequently, some somatic embryogenesis has been reported in some lily species, including *Lilium ledebourri*, *Lilium longiflorum* and *Lilium martagon*. Among plant growth regulators, picloram was the most effective hormone in SE. Transverse slices of *Lilium longiflorum* bulbs were used as explants for a thin-layer culture in MS basal medium supplemented with naphthalene acetic acid (NAA) and thidiazuron (TDZ), and it was found that somatic embryos could be directly induced [23]. *Lilium longiflorum* exosomes can be used to directly induce somatic embryos with picloram [24]. SE occurs primarily through the endogenous pathway, where the somatic embryo is formed from the inner cells of the embryonic callus tissue, and, secondly, SE can also occur through the exogenous pathway, where the somatic embryo develops from the superficial or sub-superficial cells of the EC. When the EC is transferred to hormone-free medium, it can develop into globular embryos, heart-shaped embryos and torpedo embryos until it finally forms a plant [25]. Secondly, lily somatic embryo germination can be achieved in light or in darkness. Lily somatic cell embryogenesis is usually efficiently induced with approximately 1 mm transverse tissue sections (tTCLs) [11]. The highest rate of embryogenesis (65.55%) was obtained from tTCL explants of small bulbs cultured for 3 months in *Lilium ledebourii* (Baker) Boiss [26]. In addition, microRNAs (miRNAs) play an important role in regulating gene networks involved in plant somatic embryogenesis. The miR171 family was shown to be differentially expressed in torpedo and cotyledon embryos compared to lily spheres [27]. It has been shown that Lpu-miR171 is involved in lily somatic embryogenesis through the regulation of the SCARECROW-LIKE 6 transcription factor [28].

However, the coefficient of induction of embryonic callus tissue remains low [29]. The understanding of the mechanisms of somatic cell embryogenesis formation is still in its infancy, and SE remains the least understood mode of regeneration. During somatic embryogenesis, in addition to the biochemical and morphological changes that occur, the induced tissue is closely associated with changes in gene expression throughout development [30]. The main genes involved in the initial step of early SE are LEAFY COTYLEDON1, LEC1/LEC1-LIKE (L1L), ABSCISIC ACID INSENSIVE 3 (ABI3), FUSCA3 (FUS3) and LEC2 [31]. Among the genes involved in the induction of somatic embryogenesis, the somatic embryogenesis receptor kinase (SERK) gene is thought to play an important role. The SERK gene was first isolated from carrot embryogenic cells and was often referred to as a molecular marker of somatic embryogenesis [32]. It encodes leucine-rich repeat sequence (LRR) receptor-like kinases (RLKs) with extracellular LRR structural domains, transmembrane structural domains and intracellular kinase structural domains [33]. Secondly, SERK, as a co-receptor of several receptors, remains at the centre of the study of RLK action distinct signalling in cellular activities. All SERKs contain a relatively short extracellular structural domain with five LRRs [34]. SERKs belong to the group II LRR-RLKs. In *Arabidopsis*, there are five members [35], which are named SERK1, SERK2, SERK3/BRI1-related kinase 1 (BAK1), SERK4/BAK1-like kinase 1 (BKK1) and SERK5 [36]. Arabidopsis SERK1 is expressed during syncytium and early embryogenesis [37]. Currently, SERK-like kinase genes are cloned from certain plant species such as *Arabidopsis thaliana*, maize, truncated stem alfalfa, rice and sunflower [38]. In monocots, the expression of the maize SERK gene family has been linked to the induction of embryogenesis [39]. In rice, OsSERK1 was detected at high levels of expression in callus tissues during somatic embryogenesis, suggesting that the SERK gene plays a role in mediating somatic embryogenesis [40]. The SERK gene (StSERK1) was cloned and identified in potato, and was very similar to other plant SERKs and clustered with members of the SERK family in the context of somatic embryogenesis [41].

Although SERKs have been shown to be involved in many plant activities, including male gametophyte development, immune response, etc., there are few studies on SERK genes in lily, and, in particular, there is no literature on SERK regulation of somatic embryogenesis. In this study, we cloned and identified a *LdSERK1* gene with a full CDS length of 1875 bp, encoding 624 amino acids, containing a signal peptide and transmembrane structure through transcriptomic analysis of four stages of somatic embryogenesis in Lanzhou lilies. The results of the phylogenetic tree showed that LdSERK is most closely related to rice *OsSERK1*, hence the name *LdSERK1*. The results of quantitative expression analysis showed that its expression varied in an M-shaped trend at the four stages. Meanwhile, subcellular localisation results in the onion epidermis showed that *LdSERK1* is localised to the cell membrane. In addition, we applied the virus-induced gene-silencing technique to scales, and the results showed that LdSERK1 is involved in Lanzhou lily somatic cell embryogenesis and plays a role in the induction of callus tissue in scales. Validation of the expression of this gene in *Arabidopsis* showed that *LdSERK1* enhances the ability of *Arabidopsis* seed to form callus tissue, further demonstrating its role in somatic embryogenesis.

## 2. Results

### 2.1. Somatic Embryo Induction

We mainly used Lilium davidii’s filaments as explants, and used picloram to induce callus tissue under dark culture conditions, and then induced somatic embryogenesis at different stages by 2,4-D (Figure 1A–D).

### 2.2. Differentially Expressed Genes and Analysis of Somatic Embryogenesis Receptor-like Kinase Transcript Level

To provide a global view of gene expression at the four stages of a somatic embryo, heat maps were generated using fragments per kilobase million (FPKM) values (Figure 2A). As can be seen from the heat map, there was a significant difference in expression between the four stages of the differential genes. We used three differential genes of the brassinolide pathway, LdSERK1, LdSERK2 and LdSERK3. We analysed their differential expression at four periods and found that they all varied during somatic embryo development, with LdSERK1 fluctuating more significantly (Figure 2B). KEGG pathway gene annotations indicated that cellular processes, environmental information processing, genetic information processing, human diseases, metabolism and organismal systems are mainly involved in somatic embryo development (Figure 2C). Venn diagrams of the five major databases showed that NR and KEGG were enriched with the highest number of differential genes, 15,011 and 14,972, respectively, whereas KOG was enriched with the lowest number, 9920 (Figure 2D). Furthermore, the results of the two-by-two comparison of the differential genes at each stage showed that the highest concentration of differential genes was found between HE and TE and GE and TE, with five and eight genes, respectively (Figure 2E).

### 2.3. Gene Clone and Bioinformatics Analysis

To elucidate the role of the *LdSERK1* gene in the somatic embryo brassinolide pathway, we performed gene cloning and functional analysis. The full-length CDS sequence of *LdSERK1* was screened against the Lanzhou lily transcriptome database, and the results of PCR amplification and sequencing using somatic embryo cDNA as a template showed that the full-length CDS of *LdSERK1* was 1875 bp (Figure 3A), encoding 624 amino acids, which was fully compatible with the sequence in the transcriptome. MEGA6.0 software was used to construct SERK phylogenetic trees of different plants, and the results showed that *LdSERK1* was most closely related to rice *OsSERK1* (Figure 3B). It is clustered with SERK proteins from rice, maize, pineapple and other monocotyledonous plants. Prediction of the transmembrane structure of the *LdSERK1* protein showed that the protein contained a transmembrane structure and belonged to the class of transmembrane proteins (Figure 3C). NCBI structural domain analysis revealed that the *LdSERK1* protein contains conserved structural domains such as the leucine repeat sequence region (LRR) PLN00113 and the kinase structural domain PKc_like (Figure 3D).

### 2.4. Subcellular Localisation of LdSERK1

To further understand the subcellular localisation of the *LdSERK1* protein, we constructed a fusion-expressed protein of *LdSERK1* with GFP in pCAMBIA1302 after removal of the stop codon. The pCAMBIA1302 plasmid itself was used as a control, and onion was used as the experimental material. Subcellular localisation results showed that the *LdSERK1* protein was predominantly expressed in the membrane of onion epidermal cells, in contrast to the 35:GFP control, which was expressed in the cell membrane, nucleus and cytoplasm (Figure 4).

### 2.5. Virus-Induced Gene Silencing in Lilium Scales

To determine the function of *LdSERK1* in somatic embryo development, we used virus-induced gene silencing (VIGS) to knock out *LdSERK1* in Langzhou lily. A 288 bp sequence on the *LdSERK1* cDNA was amplified with specific primers (Appendix A) and inserted into the TRV2 vector for VIGS experiments. *Agrobacterium* of TRV1:TRV2 and TRV1:TRV2-LdSERK1 was mixed in a 1:1 ratio to infect lily scales, and the null group was used as a negative control. Forty-five days after infection, we found that the scales of infected TRV1:TRV2 could produce large areas of callus (Figure 5A), whereas under infected TRV1:TRV2-LDSERK1, fewer scales produced callus and more scales were browned (Figure 5B). To confirm that *Agrobacterium* infection had occurred, we first determined the 250 bp RNA transcript of CP and MP from both TRV1:TRV2 and TRV1:TRV2-LdSERK1 basal scales of the wound of new callus (Figure 5C). In addition, the results for the detection of *LdSERK1* gene expression levels in somatic embryos showed that the expression levels of TRV2 (TK) did not change much compared to the untreated control (CK), whereas the expressions of the genes silenced by TRV-*LdSERK1* (TS) were all down-regulated (Figure 5D).

### 2.6. Phenotypic Analysis and Induction of Seed Callus Tissue in T2-Generation Arabidopsis

The *LdSERK1* gene was inserted into the pCXUN vector and placed under the CaMV35S promoter. Wild-type *Arabidopsis thaliana* (Col-0 ecotype) was transformed, and three independent transgenic lines of T2-generation transgenic lines were screened for the presence and expression of the *LdSERK1* gene (Figure 6A). The phenotypic observations showed that the transgenic positive lines were taller and had more tillers, thicker stems, more pods and larger flowers than the wild type after 20 d of growth. The qRT-PCR result showed that the *LdSERK1* gene was more highly expressed than in the wild type in three individual lines (Figure 6B). In addition, molecular detection of the hygromycin gene showed that the hygromycin gene was also amplified in three lines (Figure 6C). Seeds of wild-type Col 0 and a mixture of three independent homozygous transgenic lines (OE1, OE2 and OE3) were grown on MS solid medium supplemented with 2 mg/L 2,4-D. Yellow, granular and friable callus tissue was induced in *Arabidopsis thaliana* seeds. After 30 d of incubation, the callus tissue induction rate was 96% and 88% for transgenic and wild-type seeds, respectively, and the callus tissue particles were significantly larger than those of the wild type (Figure 6D,E). The results showed that the overexpression of the *LdSERK1* gene in *Arabidopsis* promoted the induction of callus tissue in the seeds.

## 3. Discussion

The somatic embryogenesis receptor-like kinase (SERK) gene normally plays an important role in the induction of somatic and syncytial embryogenesis in plants, and it encodes an LRR-containing receptor-like kinase protein [42]. In *Arabidopsis*, analysis of mass spectrometry studies of protein–protein interactions suggested that the SERK-1 protein is involved in the brassinolide signalling pathway [43]. *SERK* expression was detected in roots, leaves, ovules, somatic embryos, anthers and seedlings of *Arabidopsis* [44], rice [45] and maize [46]. Such a finding suggests an additional role for the SERK gene in plants beyond that associated with somatic embryogenesis. In our work, we observed that suppressing the expression of the *LdSERK1* gene reduced the rate of induction of callus tissue and somatic embryogenesis on the scales. Three transgenic lines overexpressing in *Arabidopsis* also showed that 2,4-D induced more granulated callus tissue in cotyledons. Similar results were found in lettuce transgenic lines, where the ability of seeds to form somatic embryo structures was reduced in the absence of endogenous SERK expression [42].

Understanding the regulation and control mechanisms of somatic embryogenesis is therefore a key issue in plant biology. SE is the ability of somatic cells to alter their genetic programme to produce embryogenic cells capable of giving rise to viable somatic embryos [47]. Plants can normally form embryos without meiosis and fertilisation, and SE can occur spontaneously or under certain environmental conditions in some species [48]. In nature, the SE process occurs through a variety of pathways, including the production of small bipolar structures in plant leaves, fusionless reproductive embryogenesis and micropore formation in embryos [49]. Compared to organogenesis, somatic embryogenesis has an overwhelming advantage. SE potential depends on the genotype, the exosome and the medium composition, especially the plant growth regulators (PGRs) and the environmental conditions [50]. Most lily SEs are induced using MS medium supplemented with sucrose as a carbon source, in addition to growth factors commonly used to induce embryonic callus tissue and proliferation, including NAA, 2,4-D, picloram, TDZ, kin and 6-BA [11,22]. In this study, four periods of somatic embryogenesis were induced in *Lilium davidii* var. *Unicolor* using filaments as explants and picloram and NAA, TDZ and 2,4-D as exogenous hormones (Figure 1A–D).

We successfully cloned *LdSERK1* from three SERK genes in the KEGG enrichment of transcriptomics with a full-length cDNA of 1875 bp. The *LdSERK1* protein is localised to the cell membrane upon expression (Figure 4A). The expression of *LdSERK1* varied at different times of somatic embryogenesis and at different stages of somatic embryo development. It was shown that its expression was increased at 30 d of embryonic callus (EC). The expression of *LdSERK1* gradually increased from globular embryo (GE) and heart-shaped embryo (HE) to torpedo embryo (TE), with the highest expression in TE, and then was down-regulated in cotyledonary embryo (CE) (Appendix A). Second, by analysing the transcriptomic data, we showed that the number of differentially expressed genes differed in the four stages of somatic embryogenesis. The largest distribution of up-regulated genes was at the TE stage, with 1946, and the largest number of down-regulated genes was at the HE stage, with 1013 (Appendix A).

VIGS technology has been widely used in plants to analyse gene function [51]. It is also suitable for high-throughput functional genomics. Previous studies have shown that the CMV-HL strain induces gene silencing in the lily [52]. Virus-induced gene silencing (VIGS) of LiNAC100 reduces the expression of the linalool synthase gene (LiLiS), significantly inhibiting linalool synthesis [53]. We used lily scales as explants, vacuum infiltrated with 0.8 KPA for 15 min and 0.6 KPA for 5 min, and concluded from phenotypic observations that *LdSERK1* knockdown had successfully led to a reduction in somatic embryo development (Figure 5A,B). In addition, we have analysed the status of the *Agrobacterium*-infected scales in terms of resistance screening. It was shown that the lowest browning rate of 13.33% in scales was observed at a Kan concentration of 60 mg/L (Appendix A), while the lowest browning rate of 6.67% was observed at 300 mg/L when Cef was used as a resistance (Appendix A). A very critical step in the success of scale VIGS is to control the level of browning during the infestation. Thus, our scale VIGS system suggests that it can be used for the study of molecular regulatory mechanisms of genes related to scale development in *Lilium*.

## 4. Materials and Methods

### 4.1. Plant Materials and Somatic Embryo Induction

We used Lanzhou lily as the material with flower buds about 3 cm in length at the flowering stage. Under aseptic conditions, they were disinfected with 75% alcohol for 30 s and with 2% NaClO for 16 min and were rinsed six times with sterile water. After blow-drying, the filaments were removed and inoculated into picloram 2.0 mg/L + NAA 0.2 mg/L + TDZ 0.1mg/L + Sucrose 30 g/L MS medium and incubated in the dark at 25 ℃ for 50 d. Yellowish and crisp callus tissues were induced in embryonic cultures of different stages of Lanzhou lily somatic cell embryos under 2,4-D 0.5 mg/L + 6-BA 0.2 mg/L + sucrose 30 g/L MS medium. The somatic embryo underwent four developmental stages: globular embryo, heart-shaped embryo, torpedo embryo and cotyledonary embryo.

### 4.2. Observation of Paraffin Sections and Tissue Staining

Cultures of different stages of Lanzhou lily somatic cell embryos were taken into 2 mL centrifuge tubes, and 1 mL of FAA fixative (90% of 50% ethanol + 5% glacial acetic acid + 5% formalin) was added and quickly vacuumed to ensure that the samples were completely immersed in the fixative. The samples were fixed for 2 d at room temperature. After fixation, the samples were dehydrated in different gradients of ethanol (70%, 80%, 95%, 95%, 100%) for 1 h and then transferred to a mixture of alcohol:xylene in different volume ratios (xylene:alcohol = 2:1, 1:1, 1:2) for 1 h. After treatment, they were embedded in melted blocks of wax, and the embedded blocks of wax were sliced to a thickness of 8 μm using a slicer. They were then attached to slides containing Haup’s paste and placed in a 37 °C incubator for spreading. Slides at the end of the spreading were dewaxed and stained [54]. Finally, 1–2 drops of neutral resin were added to the slide to seal the section, and the coverslip was covered and gently pressed to exhaust to produce conventional paraffin sections, which were observed and photographed under a light microscope.

### 4.3. RNA Extraction

Samples collected from somatic embryos at four developmental stages were crushed with liquid nitrogen, and total RNA was extracted according to the manufacturer’s protocol (TIANGEN, Beijing, China, code: DP432). Extracted RNA spotted at 1 μL was electrophoresed on a 1% agarose gel and checked for degradation. The purity and integrity of the RNA were also tested using a Nanodrop instrument (Nanodrop one, ThermoFisher, Waltham, MA, USA). All qualified RNA samples were snap frozen in liquid nitrogen at a low temperature of −80 °C as a back-up.

### 4.4. RNA-Seq Library Preparation and Sequencing

Library construction was performed using QC-qualified RNA. Firstly, total RNA from the samples was treated with oligo (dT) magnetic beads to enrich for polyA-containing mRNA. Secondly, the cDNA was synthesised using the SMARTer PCR cDNA Synthesis Kit. The synthesised cDNA was subjected to PCR amplification and cyclically optimised to find the best conditions for PCR amplification. The magnetic beads were used to screen the fragments for large-scale PCR amplification to obtain a sufficient amount of cDNA. Finally, damage repair, undetermined repair of full length cDNAs and ligation of SMRT junctions were used to construct a full-length library. The cDNA sequences with unligated junctions at both ends were eliminated. Primer-bound DNA polymerase was used to construct the SMRT bell library. The quality of the library was checked, and the qualified library was sequenced using PacBio RSII. The transcriptome sequencing was entrusted to Beijing Novozymes Technology Co., Beijing, China.

### 4.5. Differentially Expressed Gene Analysis

Differential expression analyses were performed for each of the different developmental stages of the samples and groups. Analyses were based on three biological replicates per stage. For KEGG pathway enrichment and GO functional analysis, significant differential genes were identified for ≥2-fold genes with a false discovery rate (FDR) < 0.05.

### 4.6. QRT-PCR Verification

LdSERK1 gene was selected for quantitative RT-PCR analysis, and gene expression and relative expression of different genes were compared by one-way ANOVA and Tukey test analysis. The statistical significance was *p* < 0.05.

### 4.7. Cloning of the LdSERK1 Gene

We used the mixed cDNAs of four stages of Lanzhou lily somatic embryos as the template for amplification. The transcribed cDNA sequence of SERK in the transcriptome data was used as a reference. A pair of specific primers, LdF1 and LdR1, was designed using Primer 6.0, and was employed to amplify LdSERK1 fragments. The LdSERK1 gene coding sequence (CDS) cDNA was cloned by RT-PCR. PCR amplification was performed using Phanta HS Super-Fidelity DNA Polymerase (Vazyme Technology Co., Ltd., Nanijing, China) with a total volume of 25 μL. The PCR reaction conditions used were as follows: initial denaturation at 95 °C for 3 min; 35 cycles of denaturation at 95 °C for 15 s; annealing at 54 °C for 15 s; extension at 72 °C for 2 min; and a final extension of 72 °C for 5 min. Gene fragments were purified using TIANGEN PCR purification test kit, cloned in pGXT vector. Transformation of *E. coli* DH5α and screening of positive monoclonal cells for sequencing were carried out. A 1875 bp putative LdSERK1 fragment was generated.

### 4.8. Vector Construction and Agrobacterium Transformation

The VIGS assay used pTRV1 and pTRV2 vectors, which have been described previously [55]. A 288 bp fragment was amplified using two primers of LdF3 and LdR3 and the cDNA of a somatic embryo as a template. This empty vector TRV2 was linearised with *EcoR*I and *Hin*dIII, and the partial fragment of LdSERK1 gene inserted into TRV2 was transformed into the *Agrobacterium* GV3101 strain. Furthermore, the pCXUN and pCAMBIA1302 plasmid were digested with *Bam*HI and *Spe*I. This full-length fragment PCR of the LdSERK1 gene by primers of LdF2 and LdR2 was seamlessly cloned into the vectors (Vazyme, Technology Co., Ltd., Nanijing, China), which were named pCXUN-LdSERK1 and pCAMBIA1302-LdSERK1.

### 4.9. Establishment of the Virus-Induced Gene-Silencing System in Lily Scales

The TRV and TRV2-LdSERK1 clone of *Agrobacterium tumefaciens* were isolated and grown in the dark at 250 rpm at 28 °C until the OD_600_ of the bacterial solution was approximately 0.8 and then collected by centrifugation at 6000 rpm for 8 min. A suspension (10 mM MgCl_2_, 200 µM acetosyringone and 10 mM MES ethane sulfonic acid) was used and resuspended to an OD_600_ of 0.8 with pH 5.6, and TRV1, TRV2 and TRV2-LdSERK1 were mixed in equal volumes (1:1) and allowed to stand in the dark for 3 h before transformation by vacuum osmosis infiltration. The group infected with the *Agrobacterium tumefaciens* mixture of TRV1 and TRV2 was used as a control. Each treatment was inoculated with 32 mid-lower scales of Lanzhou lilies, and three replicates were set up. Scales were cut to 1 cm^2^ prior to infestation and pre-cultured for 3 d in MS medium supplemented with 1.5 mg/L PIC and 0.2 mg/L NAA. The scales were vacuum infiltrated in the infiltration solution at a pressure of 0.8 KPA for 15 min, deflated for 15 min, gently shaken for 5 min, then vacuum infiltrated at 0.6 KPA for 5 min and deflated for 5 min. After infestation, the scales were blown dry and transferred to co-culture plant medium (MS + 1.5 mg/L PIC + 0.2 mg/L NAA + 100 μmol/L AS + 60 g/L sucrose + 6.5 g/L agar) and incubated in the dark at 26 °C for 3 d. After washing and transfer to I and II medium (I: MS + 1.5 mg/L PIC + 0.2 mg/L NAA + 400 mg/Lcef + 120 mg/L kan^+^ +30 g/L sucrose + 6.5 g/L agar, II: MS + 1.5 mg/L PIC + 0.2 mg/L NAA + 400 mg/L cef + 60 mg/L kan^+^ + 30 g/L sucrose + 6.5 g/L agar) in sequential dark culture for 10 d and 15 d, finally, it was transferred to induction medium (MS + 1.5 mg/L PIC + 0.2 mg/L NAA + 30 g/L sucrose + 6.5 g/L agar), and the culture was continued at 26 °C until the scales grew healing tissue and somatic embryos.

### 4.10. Subcellular Localisation

For subcellular localisation, we chose to use the onion as the material. The pCAMBIA1302 and pCAMBIA1302-LdSERK1 *Agrobacterium* clones were isolated and cultured in the dark at 28 °C, 250 rpm, until the OD_600_ of the bacterial solution was approximately 0.8, and the supernatant was discarded after centrifugation. The bacteria were washed with MS liquid resuspension (10 mM MgCl_2_ + 200 µM AS + 3% sucrose) and centrifuged, the supernatant was discarded and the OD_600_ was adjusted to 0.8 and incubated for 3 h in the dark at 28 °C. Fresh onions were selected, and onion bulbs were cut with a sterile paring knife. Then, the inner epidermis of the fresh and fleshy scales inside the bulb was injected with a syringe containing a needle, and the onions were placed at 28 °C for 48 h and observed by laser confocal microscopy.

### 4.11. Overexpression Vectors for Arabidopsis Transformation

The *Arabidopsis thaliana* flower dip method was used for infestation. The pCXUN and pCXUN-LdSERK1 from *Agrobacterium* spp. were incubated at 28 °C, 250 rpm, and centrifuged, and the supernatant was discarded. It was then resuspended in 5% sucrose and 0.02% SilwetL-77 to an OD_600_ of 0.8. Finally, it was incubated in the dark at 28 °C for 3 h. For infestation, inflorescences were immersed in the infestation solution for 30 s. After infestation, excess infestation solution was gently pipetted onto sterile paper and incubated in the dark for 24 h. Subsequently, it was grown under light/dark conditions of 16 h/8 h. T_0_-generation harvested seeds were screened by hygromycin to obtain T_1_- and T_2_-generation seeds until pure T_3_-generation seeds were obtained.

### 4.12. Quantitative Real-Time PCR Analysis

The qRT-PCR was completed using a Q3 real-time PCR system (Thermo Fisher Scineitific, Waltham, MA, USA). The system has 10 µL volume; each reaction contains 5 μL of 2 × Taq Pro Universal SYBR qPCR Master Mix (Vazyme, Nanjing, China), 0.5 μL of primers (10 μM), 1 μL of cDNA template and added ddH2O up to 10 μL. The procedure was completed according to the Q3 Semi-Quantitative PCR Operation Manual. The *Lilium* × *formolongi* EF-1a was used as an internal reference [55]. Quantitative PCR expression levels were calculated according to the 2^−ΔΔCT^ method. All experiments were performed with three biological replicates and three technical replicates. All primers in this study are shown in Appendix A.

## 5. Conclusions

Lanzhou lily is a Chinese speciality of dual-use vegetable and food varieties, and its market industrialisation value is quite high. At present, there is a lack of research into the molecular mechanisms of SE in lily, and the possibility of modifying lily SE-related genes needs to be further explored. Using traditional tissue culture, we have established a system to generate different stages of somatic cell embryos in Lanzhou lilies. This study focused on the molecular regulation of lily somatic embryogenesis by the SERK gene of the brassinolide signalling pathway based on KEGG enrichment data of the somatic embryo transcriptome. We first cloned the LdSERK1 gene associated with somatic embryogenesis in lily. In addition, we used LdSERK1 as a target gene to establish the VIGS system in Lanzhou lily for the first time and successfully reduced the expression of the LdSERK1 gene and inhibited somatic embryogenesis. Using the model plant *Arabidopsis thaliana*, the results of callus tissue induction from seeds of progeny overexpressing the LdSERK1 gene further showed that overexpression of the LdSERK1 gene promotes the development of somatic embryos. All these results positively proved that the LdSERK1 gene on the brassinolide pathway in Lanzhou lilies can promote somatic embryo formation, and, in the future, we can try to apply it to lilies with functional gene fusion constructs to improve the transformation rate of the genetic system. In conclusion, this study will provide scientific guidance for the future propagation of lily bulbs and the induction of somatic embryos.

## Figures and Tables

**Figure 1 plants-13-01495-f001:**
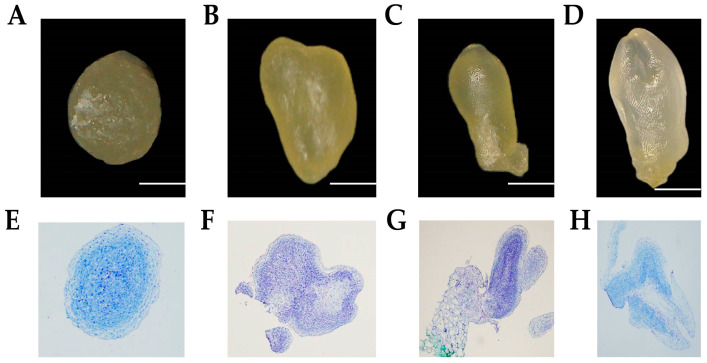
Morphological observations of Lanzhou lily somatic embryo at different stages of development. (**A**) Globular stage under the autopsy microscope. (**B**) Heart-shaped stage under the autopsy microscope. (**C**) Torpedo-shaped stage under the autopsy microscope. (**D**) Cotyledon stage under the autopsy microscope. (**E**) Globular stage under tissue staining. (**F**) Heart-shaped stage under tissue staining. (**G**) Torpedo-shaped stage under tissue staining. (**H**) Cotyledon stage under tissue staining. Scale = 1 mm.

**Figure 2 plants-13-01495-f002:**
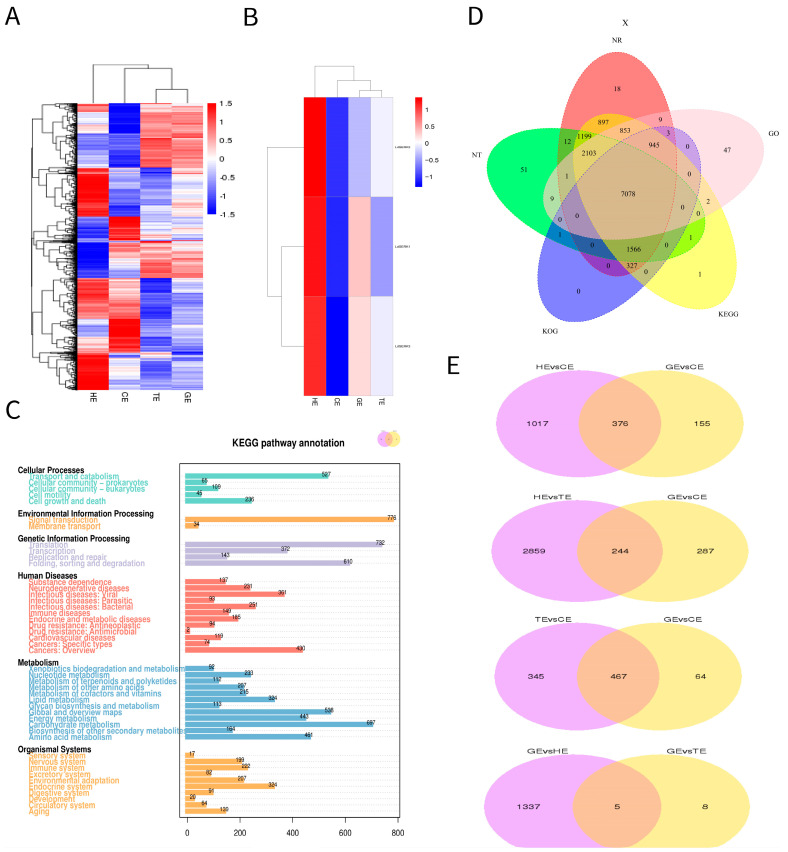
Transcriptional profiling of genes across four stages of development in somatic embryos. (**A**) Hierarchical clustering of all differentially expressed genes (DEGs) based on z−score normalised FPKM values. Blue indicates lower expression, and red indicates higher expression. (**B**) Differential expression of LdSERK1, LdSERK2 and LdSERK3 at four stages of somatic embryo development. (**C**) Distribution of differentially expressed genes in the KEGG pathway. (**D**) Gene function annotated Venn diagrams for NR, NT, KOG, KEGG and GO databases. (**E**) Genes differentially expressed between two random stages.

**Figure 3 plants-13-01495-f003:**
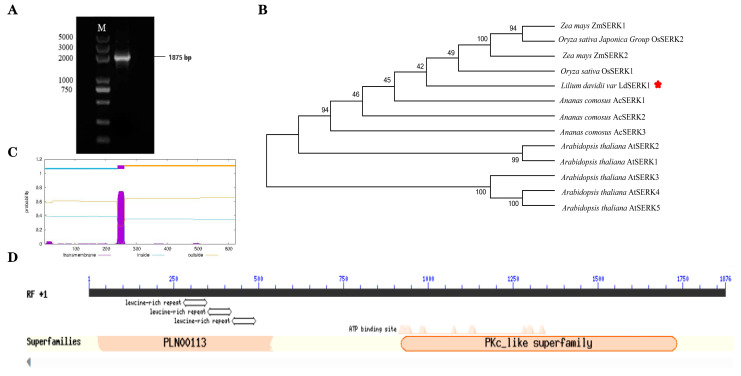
Analyses of the phylogenetic relationship, amino acid structure and LdSERK1 gene clone from lily. (**A**) Agarose gel showing the RT-PCR amplification of LdSERK1 coding DNA sequence (CDS) from lily. (**B**) Phylogenetic tree of LdSERK1 family proteins from selected plant species. The phylogenetic tree was constructed in MEGA6.0 software using the neighbour-joining method. (**C**) Prediction of the transmembrane structure of the LdSERK1 protein. (**D**) Motif analysis of the LdSERK1 protein using the NCBI domain. * It stands for the name of the genus of the species.

**Figure 4 plants-13-01495-f004:**
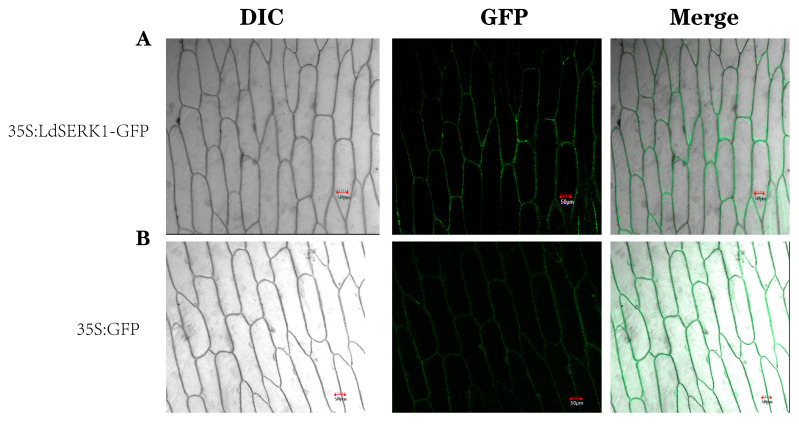
Subcellular localisation of *LdSERK1* protein in onion. (**A**) Subcellular localisation of the GFP protein in onion epidermal cells, scale bar = 50 μm. (**B**) Subcellular localisation of the *LdSERK1* protein in onion epidermal cells, scale bar = 50 μm.

**Figure 5 plants-13-01495-f005:**
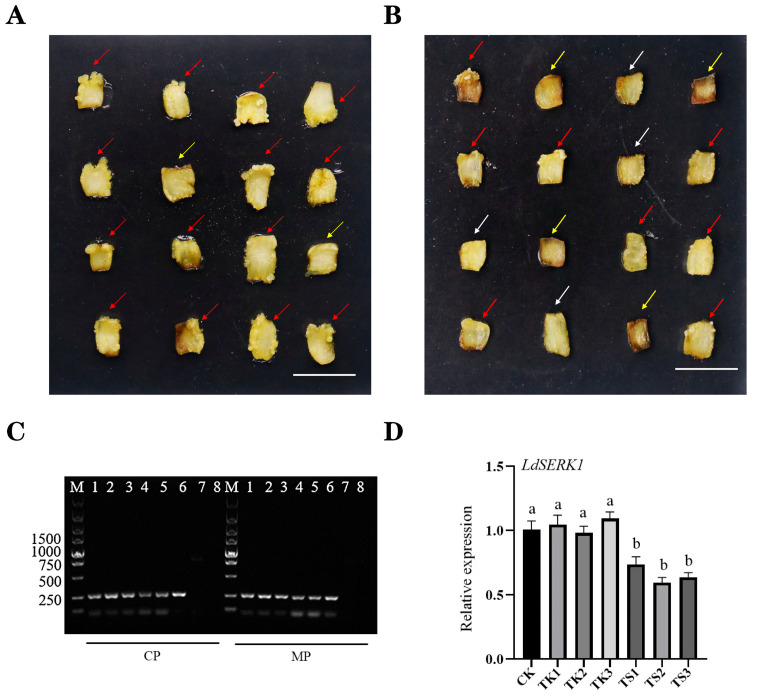
Establishment of a scale VIGS system for the *LdSERK1* gene and analysis of the assay. (**A**) Callus status after 45 days of mixed infection with TRV1 and TRV2. (**B**) Callus status after 45 days of mixed infection with TRV1 and TRV2-*LdSERK1*. Red arrows indicate a large area of healing tissue growth, yellow arrows indicate no healing tissue growth and white arrows indicate a small area of healing tissue growth. (**C**) CP and MP on TRV1 and TRV2/TRV2-*LdSERK1* backbones in nascent callus tissues of both treatment groups were detected as 250 bp transcript segments. CP: coat protein, MP: movement protein. 1–3 refer to TRV1/TRV2, 4–6 refer to TRV1/TRV2-*LdSERK1*, 7–8 refer to ddH_2_O and negative control. (**D**) RT-QPCR analysis of *LdSERK1* gene expression in control (CK), negative (TK) and positive (TS). Lowercase letters indicate result of ANOVA (*p* > 0.05).

**Figure 6 plants-13-01495-f006:**
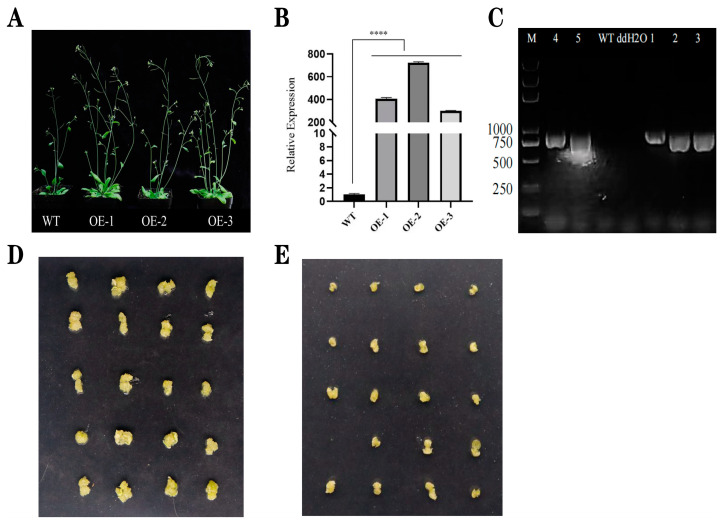
Molecular phenotype of the overexpression of the *LdSERK1* gene in *Arabidopsis thaliana* and phenotypic analysis of the induction of cotyledons into callus tissues. (**A**) Phenotypic comparison of the *LdSERK1* overexpression transgene and the wild type. (**B**) Differential expression of *LdSERK1* in overexpression and wild-type lines by fluorescence quantitative PCR. (**C**) PCR detection of Hyg in overexpression and wild-type lines. (**D**) Callus tissue induced from seeds in overexpression lines using 2 mg/L 2,4-D. (**E**) Callus tissue induced from seeds in the wild-type lines using 2 mg/L 2,4-D. **** It represents significant differences (*p* < 0.01).

## Data Availability

Data are contained within the article and Appendix A.

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
