# Peer review of "Characterisation and Expression Analysis of LdSERK1, a Somatic Embryogenesis Gene in *Lilium davidii* var. *unicolor"

_plants, 2024, doi:10.3390/plants13111495_

Round 1
Reviewer 1 Report
Comments and Suggestions for Authors
Comments
Comments and Suggestions for Authors
Dear Author,
It is my pleasure to review the manuscript entitled “Characterization and expression analysis of LdSERK1, a somatic embryogenesis gene in Lilium davidii var. unicolor” a research article submitted to MDPI Journal, Plants. Authors of this manuscript identified and characterized a number of LdSERK1 genes here. Authors have also characterized expression patterns of those genes at various developmental stages of somatic embryo through a series of bioinformatic and lab experiments. Overall, the experiments, they performed, are well and the results are convincing. Thus, the presented results take up an important topic consistent with the profile of the Journal.
However, I have some suggestions, which might improve the manuscript to make important to the wider readers.
· Improvement in English is necessary for clear understanding
· Introduction should be more constructive with rationale of the study. Elaborate clearly, why this research is necessary
· Please check the original PDF where I made comments
***Percent matching should be below 20

Moderate editing of English language required
Reviewer 2 Report
Comments and Suggestions for Authors
The authors provide analysis and experimental protocols for cloning new SERK genes in lilium that were shown to affect somatic embryogenesis in Lilly cultivars. The work is well documented and highlights some novel features of the cloned gene system in Lilium d. as well as Arabadopsis.
The work is well documented and the rationalization of steps demonstrating the down-regulation and over expression clearly connect the gene influence on somatic embryogenesis. A few comments for helping the presentation of this material:
1. Avoid figure legends moving across pages from figures - see Fig. 4.
2. Provide the indication that identifiers 'a' and 'b' in Fig. 5D are the result of post-Anova tests.
Reviewer 3 Report
Comments and Suggestions for Authors
The topic of the study is not new. These are my specific comments:
The "Introduction" has to be revised and more information about lily and its embryogenesis has to be added.
Figures 2 and 3 are not readable.
There is no information about primers used in this study.
There is no information about accession numbers of sequences reported in this study.
There is no available supplementary data.
The conclusions are too general.
